# Antileishmanial Activity and In Silico Molecular Docking Studies of *Malachra alceifolia* Jacq. Fractions against *Leishmania mexicana* Amastigotes

**DOI:** 10.3390/tropicalmed8020115

**Published:** 2023-02-14

**Authors:** Leonor Cervantes-Ceballos, Jairo Mercado-Camargo, Esther del Olmo-Fernández, María Luisa Serrano-García, Sara M. Robledo, Harold Gómez-Estrada

**Affiliations:** 1Grupo de Investigación en Química Orgánica Medicinal, Facultad de Ciencias Farmacéuticas, Universidad de Cartagena, Campus de Zaragocilla, Cartagena 130001, Colombia; 2Departamento de Ciencias Farmacéuticas, Área de Química Farmacéutica, Facultad de Farmacia, Centro de Enfermedades Tropicales de la Universidad de Salamanca (CIETUS), Instituto de Investigación Biomédica de Salamanca (IBSAL), Campus Miguel de Unamuno, Universidad de Salamanca, 37007 Salamanca, Spain; 3Unidad de Química Medicinal, Facultad de Farmacia, Universidad Central de Venezuela, Caracas 1040, Venezuela; 4Programa de Estudio y Control de Enfermedades Tropicales PECET, Facultad de Medicina, Universidad de Antioquia, Medellín 050010, Colombia

**Keywords:** *Malachra alceifolia* Jacq., *Leishmania mexicana*, phytoconstituents, molecular docking, antileishmanial activity

## Abstract

*Malachra alceifolia* Jacq. (family Malvaceae), known as “malva,” is a medicinal plant used as a traditional therapy in many regions of America, Africa and Asia. Traditionally, this plant is used in the form of extracts, powder and paste by populations for treating fever, stomachache, inflammation, and parasites. However, the ethnopharmacological validation of *M. alceifolia* has been scarcely researched. This study showed that the chloroform fraction (MA-IC) and subfraction (MA-24F) of the leaves of *M. alceifolia* exhibited a potential antileishmanial activity against axenic amastigotes of *Leishmania mexicana pifanoi* (MHOM/VE/60/Ltrod) and had high and moderate cytotoxic effects on the viability and morphology of macrophages RAW 264.7. This study reports, for the first time, possible terpenoid metabolites and derivatives present in *M. alceifolia* with activity against some biosynthetic pathways in *L. mexicana* amastigotes. The compounds from the subfractions MA-24F were highly active and were analyzed by gas chromatography-mass spectrometry (GC-MS) and by a molecular docking study in *L. mexicana* target protein. This study demonstrates the potential modes of interaction and the theoretical affinity energy of the metabolites episwertenol, α-amyrin and methyl commate A, which are present in the active fraction MA-24F, at allosteric sites of the pyruvate kinase, glyceraldehyde-3-phosphate dehydrogenase, triose phosphate isomerase, aldolase, phosphoglucose isomerase, transketolase, arginase and cysteine peptidases A, target proteins in some vital biosynthetic pathways were responsible for the survival of *L. mexicana*. Some phytoconstituents of *M. alceifolia* can be used for the search for potential new drugs and molecular targets for treating leishmaniases and infectious diseases. Furthermore, contributions to research and the validation and conservation of traditional knowledge of medicinal plants are needed globally.

## 1. Introduction

*Malachra alceifolia* Jacq. (malva) belongs to the family Malvaceae and is native to South, Central, Caribbean American and African countries. The plants of these species are characterized by their antiplasmodial and antibacterial properties [1]. Their leaves are traditionally used for their therapeutic potentials and are used in natural medicine; for example, a leaf-minced decoction is applied locally for inflammation and clogged nose in Colombia, while the leaves and shoots are used for treating malaria and leishmaniasis in Peru [2,3,4]. According to the World Health Organization (WHO), ancestral and traditional knowledge of the use of medicinal plants is central to the development of drugs and phytomedicines for preventing infectious diseases [5,6]. Malva is considered by the communities as a medicinal plant with ethnobotanical potential. The few pharmacological studies have been insufficient for the search for new antimicrobial drugs against infectious diseases in tropical countries, which is of interest for this research, which aims at the conservation, protection and rational use of these species [7,8].

In the group of these infectious diseases, we find leishmaniasis, one of the world’s neglected tropical diseases, considered a public health problem. Approximately 99 countries and territories are endemic for leishmaniasis. Its incidence includes 71 countries endemic for both visceral leishmaniasis (VL) and cutaneous leishmaniasis (CL), 8 countries endemic to VL only and 19 countries endemic to CL only, where high CL cases have been reported. These countries are Iran, Sudan, Brazil, Syrian, Türkiye, Saudi Arabia, Afghanistan, Peru, Costa Rica, Colombia, Ethiopia, Algeria, Morocco, Tunisia, and Pakistan. Leishmaniasis is an infection caused by the obligate intracellular protozoan Leishmania, with more than 90 sandfly species transmitting *Leishmania* parasites (Diptera: Psychodidae: Phlebotominae) [3,4]. In Colombia, this disease is endemic in almost all of the national territory, except in the San Andres islands and Bogota. The *Leishmania* species identified are *L. braziliensis, L. panamensis, L. mexicana, L. amazonensis, L. colombiensis, L. guyanensis,* and *L. infantum chagasi*. Recently, the species *L. equatoriensis* and *L. lainsoni* have been reported [9]. Importantly, the treatment of leishmaniasis is complicated by the administration of pentavalent antimonial, liposomal amphotericin B, pentamidine, paromomycin and miltefosine. These drugs generally have good efficacy but have serious side effects, such as cardiotoxic, nephrotoxic, and cytotoxic side effects, as well as the risk of developing insulin-dependent diabetes [10].

The high cost of and the need for prolonged and parenteral treatment with these drugs have reduced therapeutic compliance, stimulating the development of new options for disease control. Currently, the development of new antileishmanial compounds that are less toxic, more readily available, and affordable for people affected by *Leishmania* has become a challenge for many researchers. The degradome of *Leishmania* spp. contains proteases with 18 clans classified into 35 families that play an important role in virulence factors in *Leishmania* spp. These proteases are involved in many parasitic activities, such as tissue invasion, survival in macrophages, and the modulation of the host’s immune response, which contribute to the search for potential drug targets for treatment development [11,12,13]. However, knowing the mechanisms of pharmacological targets based on protein-ligand interactions as potential allosteric sites in the key proteins for the vitality and survival of *Leishmania* parasite has allowed them to become alternatives in the search for and development of leishmanicidal drugs from plant-derived products [14]. The search for new anti-leishmanial compounds that are less toxic, readily available, and affordable to the poor people most affected by this disease will be welcome to science. However, much remains to be done to identify the active principles of medicinal plants as sources of drugs with potent leishmanicidal activity as inhibitors of enzymes essential for replication, cell cycle regulation, or the production of virulence factors [15].

Consequently, in the current research, we report for the first time the in vitro antileishmanial activity of fractions of *M. alceifolia* against *L. mexicana* axenic amastigotes and assess their cytotoxicity. It should be noted that this study did not work experimentally in vitro with compounds isolated from *M. alceifolia* leaves but with fifteen phytoconstituents identified and characterized by qualitative phytochemical screening and gas chromatography-mass spectrometry (GC-MS), along with the molecular docking studies for determining potential allosteric sites from the crystal structures of pyruvate kinase (PDB: 1PKL), glyceraldehyde-3-phosphate dehydrogenase (GAPDH) (PDB: 1A7K), triose phosphate isomerase (PDB: 1AMK), aldolase (PDB: 1EPX), phosphoglucose isomerase (PDB: 1Q50), transketolase (PDB: 1R9J) glycolysis biosynthesis pathway proteins; arginase (PDB: 4ITY) polyamine salvage pathway protein, and cysteine peptidase A (PDB: 2C34) proteinase pathway protein of *L. mexicana,* potential therapeutic targets in the developing novel antileishmanial drugs [15].

## 2. Material and Methods

### 2.1. Chemicals

Organic solvents of p.a. grade, including n-hexane, chloroform, ethyl acetate, methanol for gas chromatography grade, and silica gel 60 GF254, for column chromatography and thin-layer chromatography (TLC)-coated plates on aluminum foil (20 × 20 cm; 200 µm were purchased (Merck, Darmstadt, Germany). For the phytochemical screening, Drangendorff, Baljet, KOH 5%, citroboric acid, ferric chloride 10%, vanillin-concentrated sulfuric acid, Lieberman–Burchard and Bornträger were purchased (Merck, Darmstadt, Germany). M199 media, (RPMI) 1640 media, sodium benzyl penicillin, HEPES, fetal bovine serum (FBS), gentamicin, L-glutamine, D-glucose, NaCl, KCl, KH_2_PO_4_, NaHCO_3_, Na_2_HPO_4_, and [3-(4,5-Dimethylthiazol-2-yl)-2,5-diphenyltetrazolium bromide] (MTT) were purchased from Sigma Chemical Co., St. Louis, MO, USA. Giemsa eosin methylene blue solution was obtained from Merck, Darmstadt, Germany.

### 2.2. Parasites and Cell Culture

The axenic amastigotes *L. mexicana pifanoi* (MHOM/VE/60/Ltrod) and RAW 264.7 murine macrophage cell line were obtained from Dr. Luis Rivas Centro de Investigaciones Biológicas Margarita Salas (C.S.I.C), Madrid, Spain.

### 2.3. Extract and Fractions Preparation

The *M. alceifolia* Jacq. leaves were collected in June and July 2019 at Arjona-Bolivar on the northern coast of Colombia (Resolution N° 27 Junie 2014, CARDIQUE). The mean annual temperature oscillates between 25 °C and 32 °C, and the relative humidity is about 80%. A voucher specimen (JBB 37103) has been deposited at the Botanical Garden Jose Celestino Mutis Bogota-Colombia. The *M. alceifolia* leaves were dried at room temperature and were then ground with a blender. A total of 3 kg of plant samples were extracted by maceration in 98% ethanol (15 L) for 4 days (1:10 *w*/*v* ratio) at room temperature. The crude extract was obtained after filtration with Whatman N°. 1 filter paper and were evaporated under low pressure below 55 °C in a rotorevaporator Heidolph model Hei-Vap precision (250 mbar). The dry extracts were evaluated by phytochemical screening. Then, 180 g of dried crude extract was subjected to column chromatography using silica gel 60 Å, 70–230 mesh, 63–200 µm as the stationary phase. The size of the column was 10 cm in diameter × 70 cm in height. The solvent mixtures were used as the mobile phase in order of increasing polarity, as follows: CHCl_3_/100:0 *v*/*v*; hexane/CHCl_3_/95:5 *v*/*v*; CHCl_3_/EtOAc/95:5 *v*/*v*; EtOAc/MeOH/95:5 *v*/*v* and MeOH/100:0 *v*/*v*. In total, twenty-four fractions were collected in 500 mL volumes and pooled in seven fractions (MA-24A at MA-24H). The qualitative detection of secondary metabolites of fractions was examined by a thin-layer chromatography (TLC) silica gel 60 GF254 plate. TLC plates were detected under UV light of 254 and 365 nm [16,17]. All the fractions were concentrated at reduced pressure and solubilized in DMSO until their use in the leishmanicidal and cytotoxic bioassays.

#### Gas Chromatography-Mass Spectrometry (GC-MS) Analysis

Agilent Gas Chromatograph 7890A series (Agilent Technologies, Inc., Santa Clara, CA, USA) was used for the identification of the major components of subfractions (MA-24F). The qualification and quantification spectral were carried out using Agilent MassHunter Qualitative Analysis (Version 10.0; Agilent Technologies, Inc., Santa Clara, CA, USA) and NIST MS Search Program v.2.3 (NIST, 2017) software program. 

### 2.4. In Vitro Anti-Amastigote Activity on L. mexicana pifanoi

The axenic amastigotes *L. mexicana pifanoi* (MHOM/VE/60/Ltrod) were grown in 199 medium supplemented with 1.25 g/L glucose, 2.5 g/L trypticase, 500 μL/L gentamicin, 0.375 g/L glutamine, 25 mg/L hemin and 10% FBS. The parasite cultures were maintained at 7–15 × 10^6^ parasites/mL in the growth medium for each stage, M199 + HIFCS at 32 °C for 4 days. We collected parasites in the late exponential growth phase (12–15 × 10^6^ cells/mL) by centrifugation at 1.000 rpm, decanted the supernatant carefully, and washed them twice in HBSS at 4 °C. We then resuspended the cells in HBSS at a final density of 4 × 10^7^ cells/mL and kept them on ice. Unless otherwise stated, these conditions were maintained for the rest of the experiments. In 1.5 mL microcentrifuge tubes, we prepared the following concentrations from the fractions and subfractions: 2.5, 10.0, and 50.0 µg/mL. In a sterile 96-well microplate, we placed 100 µL/well of the parasite suspension (4 × 10^7^ cells/mL) in HBSS. Then, 100 µL/well of the different concentrations of MA-IC and MA-24F fractions were added and mixed by pipetting; untreated parasites (control) and positive control miltefosine (30 µM) were used. The amastigotes were incubated overnight at 32 °C. The activity was measured spectrophotometrically at 595 nm using a Bio-Rad CA, USA, model 680 microplate [18,19]. The results represent the means ± SEM from 2 experiments performed in triplicate. *p* < 0.05 (*), *p* < 0.001 (**) and *p* < 0.0001 (***) compared to the control (untreated parasite).

### 2.5. Cytotoxicity Assay RAW 264.7 Cell

The cytotoxic activity of all fractions was evaluated in the RAW 264.7 murine macrophage cells by the MTT method, as described previously. In brief, cells were cultured in RPMI 1640 medium supplemented with 10% fetal bovine serum (FBS) and 1% of antibiotics (penicillin-streptomycin (10.000 U/mL) at 100.000 cells/mL. Then, 10.0 µL of the extract and fractions or DMSO were added to obtain the concentrations (2.5, 10.0 and 50.0 μg/mL). Cells cultured in the medium alone were used as a negative control (no toxicity), while cells exposed to miltefosine were a positive control (toxicity). Cells were incubated at 37 °C, 5% CO_2_ for 72 h; then, the effect of each product on the viability of cells was determined by incubating exposed and unexposed cells for 3 h with 10.0 μL of 3-(4,5-dimethylthiazol-2-yl)-2,5-diphenyltetrazolium bromide. The MTT was reduced by succinate mitochondrial dehydrogenase to purple formazan that was then solubilized with 100.0 μL/well isopropanol 50% and SDS 10%, and its concentration was determined by optical density at 570 nm in a spectrophotometer (model 680 Bio-Rad microplate CA, USA) [20,21,22]. The results represent the means ± SEM from 2 experiments performed in triplicate. *p* < 0.05 (*), *p* < 0.001 (**) and *p* < 0.0001 (***) compared to the control (untreated cells).

### 2.6. Molecular Docking

In molecular docking studies with structure-based virtual screening and docking we used various bioinformatics tools, such as AutoDock vina [23,24], BIOVIA Discovery, Studio 2020 pipeline (Accelrys Inc., San Diego, CA, USA, http://www.accelrys.com (accessed on 16 December 2022), and LigPlot [25,26]. All the 3D structures of the target proteins pyruvate kinase (PDB: 1 PKL), glyceraldehyde-3-phosphate dehydrogenase (PDB: 1A7K), triose phosphate isomerase (PDB: 1AMK), aldolase (PDB: 1EPX), phosphoglucose isomerase (PDB: 1Q50), transketolase (PDB: 1R9J), arginase (PDB: 4ITY) and cysteine peptidases A (PDB: 2C34) of *L. mexicana* were downloaded from the Protein Data Bank RCSB (PDB). The structures of compounds identified from the subfraction from *M. alceifolia* were obtained from PubChem and optimized using MMFF94 as a force field and conjugate gradient. The compounds were energy minimized, and the lowest energy conformation of each compound was used for docking studies within the allosteric domains of the evaluated proteins. Finally, the scoring function was determined [27]. 

#### 2.6.1. Prediction of Allosteric Binding Sites

Cavity-detection-guided blind docking was used to predict potential allosteric sites of the target protein of *L. mexicana*. CB-Dock and DoGSite were used to assess the features of the predicted allosteric sites [28,29,30,31]. Amphotericin B and miltefosine were also included in the study retrieved from PubChem with compound identifiers (CIDs) 5,280,965 and 3599, respectively. All ligand structures were energy minimized using the universal force field under the conjugate gradient and converted to the partial charge and atom type (pdbqt) format using Autodock Tools. The protein structures were then saved in the Protein Data Bank format (pdb) using PyMOL. The protein structures were then converted to an AutoDock Vina-compatible pdbqt format using the “make macromolecule” option in PyRx.

#### 2.6.2. Virtual Screening

AutoDock Vina were employed for the virtual screening process [23,24]. The fifteen compounds of the active fraction MA-24F determined by GC-MS were screened against eight target protein biosynthetic pathways of *L. mexicana* using a grid box dimension of 40 × 40 × 40 Å3 and center at (120.244; 106.962; 21.619) Å was specified for the pyruvate kinase; (64.631; 61.312; 147.774) Å glyceraldehyde-3-phosphate dehydrogenase; (4.001; 0.012; 4.006) Å Triose phosphate isomerase; (−95.015; −54.090; −19.013) Å aldolase; (19.007; 17.000; 135.021) Å phosphoglucose isomerase; (9.45; 10.030; 57.145) Å transketolase; (17.271; −11.540; −0.7610) Å arginase; and (−14.00; 2.000; 1.000) Å cysteine peptidase A. Compounds that possessed binding affinity energies (scoring function) more than −8.0 kcal/mol were not considered for downstream analysis. The scoring function was used to determine the site of a ligand binding to predict the binding affinity and to identify potential drug leads for a given protein target. The output docking scores were given as precision glide scores (GScore); the GlideScore is an empirical scoring function that approximates the ligand binding free energy [32,33]. The reported binding energies are the mean of six runs, and statistical uncertainties are given within one standard error of the mean.

#### 2.6.3. Characterization of Mechanism of Allosteric Binding Sites

The characterization of the mechanism of binding interactions and their types, including hydrogen bonds, alkyl, π-alkyl, halogen, and the Van der Waals interactions formed between the protein of *L. mexicana* and the ligands (major metabolites of the fraction MA-24F of *M. alceifolia*), were determined and analyzed by Discovery Studio and LigPlot + v1.4.5 using default parameters [34].

#### 2.6.4. Pharmacological Profiling

The pharmacological profiling of the fifteen compounds with significant affinity with the *L. mexicana* protein was not evaluated.

## 3. Results and Discussion

### 3.1. Phytochemical Screening for Thin-Layer Chromatography (TLC) and GC−MS Analysis of M. alceifolia Extract and Fraction

The phytoconstituents isolated from the *M. alceifolia* have been little known. This research describes for the first time the secondary metabolites present in chloroform fraction of *M. alceifolia* leaves and sub-fraction MA-24F. The secondary metabolites described by GC-MS could be responsible for the observed antileishmanial effects. The phytochemical screening by TLC indicated terpenoids/sterols and tannins in all fractions (Appendix A). The *n*-hexane fraction MA-IH was obtained a yield of 2.4%. The chloroform fraction MA-IC gave a yield of 9.8%, concentrating in it the highest number of metabolites that were soluble in the chloroform solvent, followed by the ethyl acetate fraction (MA-IA) with 4.6%. The methanol fraction (MA-IM) obtained a yield of 5.8%.

The GC–MS analysis of MA-24F active fraction allowed the identification of fifteen compounds, which were classified as fatty acid derivatives, sesquiterpenes, diterpenes, triterpenes, tocopherol, phytosterol, and others (Appendix A). The first report of these species, methyl 10,11-tetra decadienoate, santalane, 2-pentadecanone,6,10,14-trimethyl, methyl 9,10-octadecadienoate, phytol, α-tocospiro A, α-tocospiro B, γ-tocopherol, β-sitosterol acetate, stigmasterol, (24R)-stigmast-5-en-3beta-ol, methyl commate A (**1**), α-amyrin (**2**), Solanesol, and episwertenol (**3**), is shown in Figure 1.

The medical use of plant products in drug discovery and development is not surprising, as humans have been using many plant-derived materials and secondary metabolites for centuries. Plant secondary metabolites represent viable options for current treatments of infectious diseases. Notably, terpenes constitute the largest and most diverse group of phytoconstituents from diverse natural sources [35,36,37].

In this context, terpene compounds identified from *M. alceifolia* leaves contribute to the search for new plant species with potential sources of novel and selective agents for treating neglected tropical diseases, especially protozoan parasites, due to their highly selective mode of action. Revisions of the genus *Malachra* have reported the presence of terpenes in the aerial parts of the plant with little study of their biological potential [1,38]. Studies report that some terpenes possess a potential mechanism of action for treating visceral and cutaneous leishmaniasis. Among them, the plasma membrane parasites are one of the main targets in which terpenes act as spacers, increasing the membrane fluidity. This mode of action is associated with the ability of terpenes to insert themselves between the fatty chains present in the lipid bilayers from the membrane [39,40].

### 3.2. Antileishmanial and Cytotoxicity Activity of Fractions M. alceifolia axenic Amastigotes L. mexicana pifanoi and RAW 264.7 Macrophages

The amastigote life forms of *Leishmania* are nonmotile and give immunopathogenesis to the parasite [41]. Thus, the growth inhibitory effects of the fraction MA-IC and sub-fraction MA-24F *M. alceifolia* were assessed against the growing axenic amastigote *L. mexicana pifanoi* (MHOM/VE/60/Ltrod). The amastigotes tested were at the dose concentrations of 2.5, 10.0, and 50.0 μg/mL of fraction MA-IC and MA-24F, with untreated and miltefosine 30.0 µM used as drug control. The MA-IC and MA-24F fractions dose-dependently reduced axenic amastigotes with an IC_50_ value of 15.65 ± 0.74 and 5.78 ± 0.46 μg/mL, respectively, higher compared to the control (untreated parasite) and a similar behavior to the control drug (miltefosine) decreasing the percentage of parasite survival (Figure 2).

The cell cytotoxicity (CC_50_) of the fractions MA-IC and sub-fractions MA-24F were evaluated along with miltefosine, amphotericin B and untreated cells as a control on macrophage cell lines RAW 264.7 to study its safe dose. The RAW 264.7 macrophages were incubated with different concentrations of fractions (2.5, 10.0, and 50.0 μg/mL)/miltefosine (30.0 µM)/amphotericin B (20.0 µM), and the cell viability was assessed using the 3-(4,5 dimethyl-thiazol-2-yl)-2,5-diphenyltetrazolium bromide (MTT) assay. It was observed that the MA-IC fraction and the MA-24F sub-fraction have higher and moderate cytotoxic effects on the viability and morphology of the macrophages with a CC_50_ values of 47.23 ± 15.57 and 49.02 ± 8.12 μg/mL, respectively, while miltefosine and amphotericin B showed higher toxicity (data not shown) (Figure 3).

In this study, the active fractions showed an effect on the axenic amastigote of *L. mexicana pifanoi* (MHOM/VE/60/Ltrod), as well as high and moderate cytotoxicity at tested concentrations of 2.5, 10.0 and 50.0 µg/mL. This indicates the need for further work on the purification of the compounds present in these fractions, followed by evaluation by bio-directed assays of other macrophage lines.

The terpenes can cause biochemical, metabolic and molecular reactions affecting the morphology, survival and development of the *Leishmania* sp. parasite [42,43,44]. This study has demonstrated, for the first time, fractions rich in secondary metabolites such as terpenes and their derivatives present in the leaves of *M. alceifolia*, which confer a possible in vitro action against the axenic amastigote and in silico action against some key proteases in the biosynthetic pathways of *L. mexicana*. However, studies have demonstrated that triterpene extracts and fractions obtained from *Boswellia* and *Commiphora* spp., particularly methyl Commate A (boswellic acids), exhibit activity against axenic amastigotes *L. donovani* [45,46,47]. β-Amyrin from *Leuconotis eugenifolius* inhibited *L. donovani* and extracts of species of the genus *Eugenia* (*E. uniflora* and *E. umbeliflora*) against *L. amazonensis* [48]. The action of amyrins and episwertenol on *Leishmania* have been reported in only a few studies; in this work, we report, for the first time, possible actions of these compounds identified in active fractions of *M. alceifolia* leaves. Many compounds from plant sources have shown potential of antileishmanial lead activity.

However, some terpene phytoconstituents identified in other plant species can interfere with *Leishmania* DNA topoisomerase and modulation of the immune response, stimulating Th1-producing pro-inflammatory responses, the production of NO in infected macrophages, cell differentiation blockade, and cell cycle progression from G2 to M.

Phases are, among other mechanisms, responsible for parasite survival [49]. Among them, the *Acacia nilotica* compound 13-docosenoic acid, 9,12-octadecadienoic acid, lupeol and 6-octadecanoic acid, showed effective binding with the potential protein’s targets sterol 24-c-methyltransferase, trypanothione reductase, pteridine reductase, adenine phosphoribosyl transferase, and *Corchorus capsularis* β-sitosterol *L. donovani* trypanothione reductase [50]. Therefore, due to the lack of interest in the clinical evaluation studies for neglected tropical infectious diseases (NTDs), they are unlimited. Importantly, the ethnobotanical and ethnopharmacological validation of herbal medicine contributes to the ability of many communities in the world to mitigate some health problems [3,7,51].

### 3.3. Molecular Docking Studies of M. alceifolia Active Fraction Ma-24F of Major Constituents with the Potential Drug Targets of L. mexicana

The fifteen compounds of the active fraction MA-24F could be determined by GC-MS analysis compared with reference substances (data not shown) and evaluated experimentally by molecular docking with eight PDB-registered proteins as possible molecular targets that play important roles in glycolysis biosynthesis, proteinases, and polyamine salvage pathways of *L. mexicana* [15,52,53] (Appendix A). The compounds episwertenol, α-amiryn, methyl commate A, and control drugs (amphotericin B and miltefosine) were observed to dock with binding affinity estimation energies close to −8.0 kcal/mol into the allosteric sites for pyruvate kinase, glyceraldehyde-3-phosphate dehydrogenase (GAPDH), triose phosphate isomerase, aldolase, transketolase, arginase, and cysteine peptidases A (Table 1 and Appendix A).

The episwertenol compound showed an allosteric binding site on the seven proteins of *L. mexicana*, showing the highest binding affinity of −9.1 ± 0.1 kcal/mol^−1^ with GAPDH forming hydrophobic contacts with Ile13, Thr197, Ala198, Thr199 (Table 1, Figure 4). Then, pyruvate kinase, which docked into pocket 1, interacted with Ser439 via hydrogen bonding and Arg19 and Arg348 via a hydrophobic bond (Table 1, Appendix A); triose phosphate isomerase interacted via two hydrogen bonds with Arg98 and Lys112 and formed hydrophobic contacts with Ile68, Ala69, Lys70, Phe74, and Glu104 (Table 1, Appendix A). Aldolase interacted with Arg52 via one hydrogen bond and Arg313 via hydrophobic bonding (Table 1, Appendix A); transketolase interacted via one hydrogen bond with Asp53 and formed hydrophobic contacts with Phe104, Pro52, Pro107, and Arg 332 (Table 1, Appendix A). Arginase interacted via one hydrogen bond with Lys198 and formed hydrophobic contacts with Leu190, Vla193, Leu201, and His202 (Table 1, Appendix A). Additionally, cysteine peptidases A interacted via one hydrogen bond with Pro30, Phe96, and Tyr58 via hydrophobic bonding (Table 1, Appendix A). The α-amiryn compounds that showed an allosteric binding site with proteins of the glycolysis biosynthesis pathway presented the highest binding affinity of −9.9 ± 0.1 kcal/mol^−1^ with pyruvate kinase through hydrogen bonds with Ser439, hydrophobic bonds with Gln42, Leu74, Val76, Glu438, Phe463 (Table 1, Figure 5). This compound with GAPDH and aldolase proteins showed binding affinity of −8.6 ± 0.1 kcal/mol^−1^, forming hydrophobic contacts with Arg12, Ile13, Thr197, Thr199, interacted via one hydrogen bond with Asp38 (in GAPDH protein) and Glu44, Leu280, Ala312 via hydrophobic bond (in aldolase protein) (Table 1, Appendix A). α-amiryn with transketolase showed a binding affinity of −8.2 ± 0.1 kcal/mol^−1^, forming hydrophobic contacts with Trp309, Val 55, Leu313, Phe327, Val328, Met331, Arg332 (Table 1, Appendix A), and allosteric binding site with proteins of the polyamine salvage pathway, and the compound with the arginase binding affinity of −8.3 ± 0.1 kcal/mol^−1^ through one hydrogen bonds with Arg191 and Val193, Lys198, Leu201, Ala208 (Table 1, Appendix A). The methyl commate A compound interacted with pyruvate kinase forming hydrophobic contacts with Ile78, Ala347, Arg348, and Glu438 and two hydrogen bonds with Ser46 and Glu348 (Table 1, Figure 6). The compound with aldolase protein interacted via four hydrogen bonds with Asp43, Lys239, Ala312, and Arg313, and it interacted with Glu44, Leu121, Leu280, Ala312, Arg313 via hydrophobic bonds (Table 1, Appendix A). Transketolase formed hydrophobic contacts with Phe104 and Met331 via one hydrogen, interacted with Met35, Pro30, Tyr58, Pro60, and Pro95 via hydrophobic bonds (Table 1, Appendix A), and interacted with Cys56, Asp45, Lys57 of cysteine peptidases A via three hydrogen bonds (Table 1, Appendix A). A theoretical study (not discussed in the paper) describes the conformational changes that occur when interacting with the structures of the studied proteins. These partial results can provide basic information for the prediction and identification of possible amino acids present in the allosteric domains of the proteins responsible for the survival of *L. mexicana* (Table 1 and Appendix A).

The proteins of biosynthesis of glycolysis, polyamine salvage and proteinase are involved in the steps of parasite invasion and migration inside the host, as well as in immune evasion, pathogenesis and disease outcome [54]. The *L. mexicana* protein targets the 1PKL glycolytic processes, the 1EPX glycolytic pathway, the conversion of fructose-1,6-bisphosphate to glyceraldehyde-3-phosphate, and dihydroxyacetone phosphate; 1AMK plays a central role in the glycolysis process as a catalyst of dihydroxyacetone phosphate (DHAP), D-glyceraldehyde-3-phosphate (GAP). 1R9J transfers two-carbon glycolaldehyde units from ketose-donors to aldose-acceptor sugars; 1A7K and 1Q50 catalyze oxidation-reduction reactions and provide defense against oxidative stress. 2C34 is central to host-parasite interactions and virulence factors, while 4ITY is involved in cellular growth, survival and robust pathogenesis [15,55,56,57]. The cysteine peptidases proteins play a crucial role in host cell-parasite interaction, thus affecting host immune response, autophagy, cell death and growth suppressors. Glycolysis biosynthesis *Leishmania* depends on the host for the carbon sources which are used for ATP generation in peroxisomes such as cell organelles called glycosomes [58].

Amphotericin B was the most highly docked control drug in this study, with binding affinity estimation energies close to −8.0 kcal/mol^−1^ at the allosteric site of all proteins. However, it showed higher binding affinity estimation energies close to −9.0 kcal/mol^−1^ at the allosteric site of GAPDH protease, forming multiple hydrogen bonds with Ile13, Met39, Arg92, Gly112, Ala135, Ser165, Gly22, Cys166, Thr167, Thr225, Arg249, and Asn335 via hydrophobic bonding. Subsequently, pyruvate kinase formed multiple hydrogen bonds with Arg19, Ser46, Asn67, Asn432, Ser439, Glu462 and formed hydrophobic bonds with Arg22, Ile 41, Gly44, Val76, Glu438, Val440, Phe463, and Lys467 (Table 1). Miltefosine was the most-docked control drug with binding affinity estimation energies close to −4.0 kcal/mol^−1^ at the allosteric site of all proteins (Table 1). Amphotericin B (antifungal) and miltefosine (anticancer) are drugs used to treat CL and VL by binding to the ergosterol of the plasma membrane forming pores and permeabilizing the outflow of ions, water and glucose molecules through the lipid bilayer, thus causing death and autophagy, with the release of reactive oxygen species (ROS) causing cell death by apoptosis in the parasite [59,60,61].

This study reports for the first time the predicted glide dock scores (GScore) at the allosteric binding site of the target protein pyruvate kinase, GAPDH, triose phosphate isomerase, aldolase, transketolase, arginase and cysteine peptidase A in *L. mexicana* with phytocomponents identified in the Ma-24F fraction of leaves of *M. alceifolia* and the drugs amphotericin B and miltefosine used as a control for the treatment of VL and CL (Table 1 and Appendix A). The scoring function data obtained for each molecule selected as promising (α-amiryn, episwertenol and methyl commate A) were contrasted and validated with the results of the measurement of the molecular binding of these ligands using the prediction from the CB-Dock program (Appendix A) [28].

Actually, studies of the genus *Malachra* have only reported phytoconstituents with no potential activity against *Leishmania* parasites. In this work, we report for the first time the prediction of the binding mode and affinity of pyruvate kinase, glyceraldehyde-3-phosphate dehydrogenase, triose phosphate isomerase, aldolase, phosphoglucose isomerase, transketolase, arginase and cysteine peptidases A within the allosteric binding site of important protein targets in the glycolysis biosynthesis, polyamine salvage and proteinase pathways of *L. mexicana*. The empirical scoring function of the AutoDock Vina Program was widely used for pose and affinity prediction. We did not evaluate the pharmacokinetic studies of the compounds α-amyrin, episwertenol and methyl commate A, which were in the active fractions of *M. alceifolia* leaves.

## 4. Conclusions

Triterpenes isolated from plants have demonstrated potential antiprotozoal activity. This study demonstrated the possible in vitro effects of *M. alceifolia* leaf extracts and fractions against *Leishmania mexicana* and determined the binding affinities of the compounds present in these fractions with possible allosteric domains of target proteins present in the pathways of glycolysis, polyamine salvage, and proteinase biosynthesis responsible for the survival of the parasite. The study identified three potential bioactive compounds α-amiryn, episwertenol and methyl commate A with binding affinity estimation energies higher than a −8.0 kcal/mol^−1^ into allosteric sites with pyruvate kinase, GAPDH, triose phosphate isomerase, aldolase, transketolase, arginase, and cysteine peptidase A. Additionally, the effect on the axenic amastigote *L. mexicana pifanoi* (MHOM/VE/60/Ltrod) of high and moderate cytotoxicity at the tested concentrations of 2.5, 10.0 and 50.0 µg/mL leads us to continue the search for obtaining pure compounds from these plant species and their experimental evaluations in vitro and in vivo. Moreover, we were predicted to be potential molecules against *L. mexicana*, so experimental evaluation is needed to corroborate their bioactivity.

## Figures and Tables

**Figure 1 tropicalmed-08-00115-f001:**
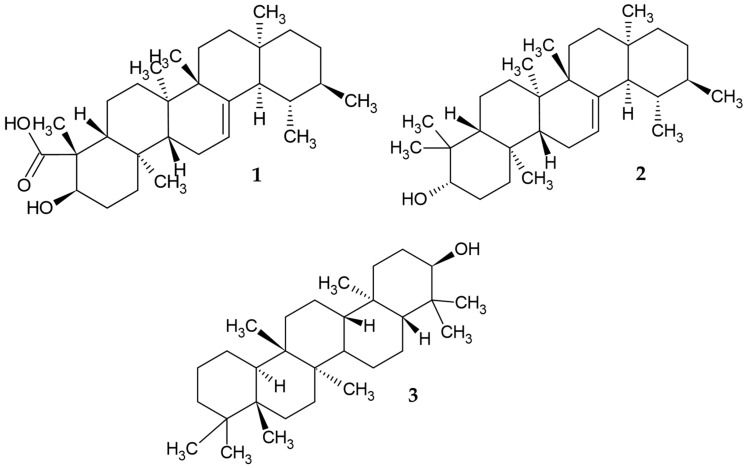
Terpenoids identified in the active fraction of MA-24F in leaves of *M. alceifolia* by GC-MS. These are the compounds with the best score protein-ligand interactions in *L. mexicana* proteins.

**Figure 2 tropicalmed-08-00115-f002:**
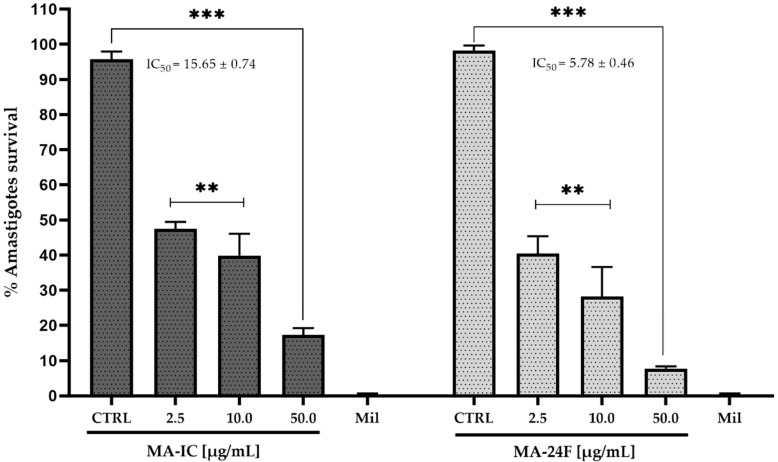
The effect of fractions *M. alceifolia* on axenic amastigotes *L. mexicana pifanoi* survival. Parasites were incubated with different concentrations (2.5, 10.0 and 50.0 µg/mL) of chloroform fraction MA-IC and sub-fractions of MA-24F, with miltefosine (positive control 30.0 µM) and untreated parasite. The amastigote viability was determined by the MTT assay. Each bar represents the mean ± standard error of at least two independent experiments, *p* < 0.001 (**) and *p* < 0.0001 (***) compared to the control (untreated parasite).

**Figure 3 tropicalmed-08-00115-f003:**
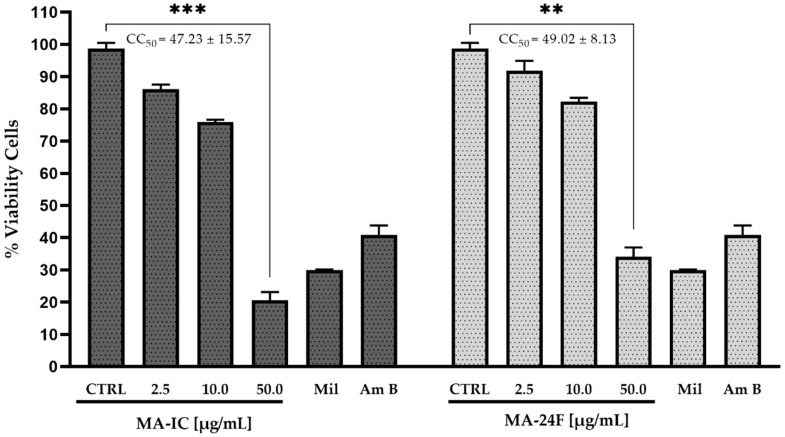
Cytotoxicity of fractions *M. alceifolia* on RAW 264.7 murine macrophage cell viability. RAW 264.7 were incubated with different concentrations (2.5, 10.0 and 50.0 µg/mL) of MA-IC and MA-24F fractions, untreated cells (control), miltefosine (30.0 µM) and amphotericin B (20.0 µM) for 72 h. The cell viability was determined by the MTT assay. Each bar represents the mean ± standard error of at least two independent experiments, *p* < 0.001 (**) and *p* < 0.0001 (***) compared to the control (untreated cells).

**Figure 4 tropicalmed-08-00115-f004:**
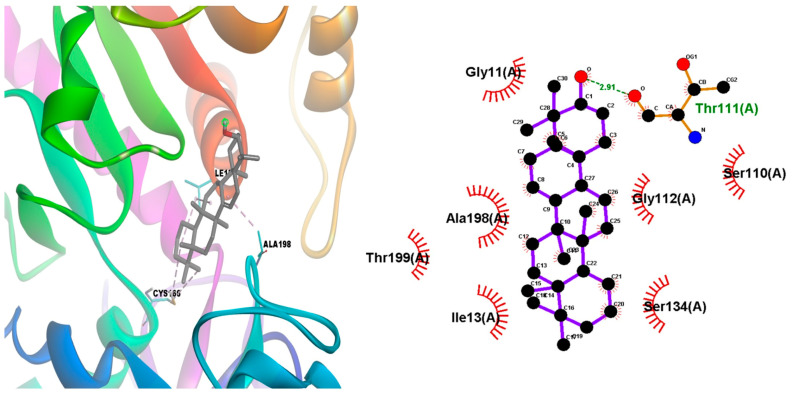
Episwertenol compound interaction of allosteric binding sites of target proteins GAPDH (PDB: 1A7K).

**Figure 5 tropicalmed-08-00115-f005:**
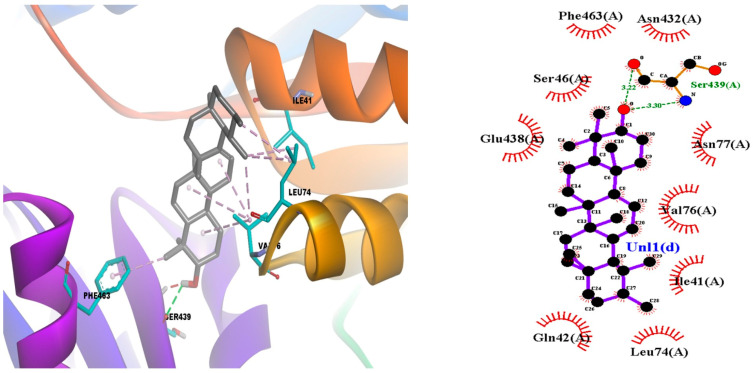
α-Amiryn compound interaction of allosteric binding sites of target proteins pyruvate kinase (PDB: 1PKL).

**Figure 6 tropicalmed-08-00115-f006:**
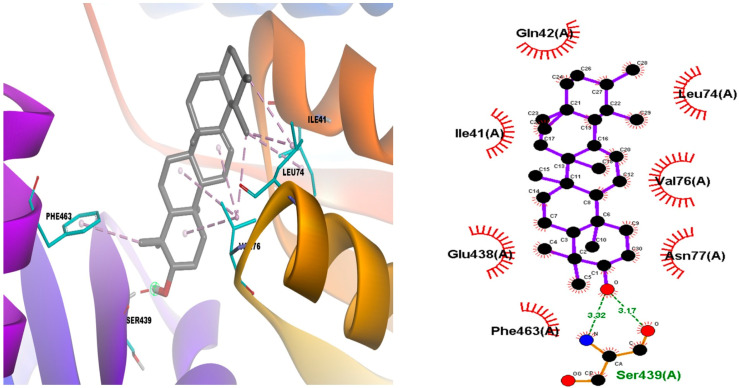
Methyl commate A compound interaction of allosteric binding sites of target proteins pyruvate kinase (PDB: 1PKL).

**Table 1 tropicalmed-08-00115-t001:** Scoring function estimation of binding affinity energy and molecular interactions of phytocomponents identified in the Ma-24F fraction of leaves of *M. alceifolia* by GC-MS upon docking in a potential allosteric sites of significant *L. mexicana* target proteins for AutoDock Vina program.

Proteases Targets *L. mexicana*	Ligand/Drugs	Allosteric GScore (Binding Energy kcal/mol^−1^)	Interacting Hydrophobic Residues	Interacting Hydrogen Bond
Glyceraldehyde-3-Phosphate Dehydrogenase (GAPDH) * PDB: 1A7K	Episwertenol	−9.1± 0.1	Ile13, Thr197, Ala198, Thr199, Cys166	Thr111
	α-amiryn	−8.6 ± 0.1	Arg12, Ile13, Thr197, Thr199, Ala135, Ala198	Asp38, Met39
	Amphotericin B	−9.3 ± 0.1	Ile13, Met39, Arg92, Gly112, Ala135, Ser165, Gly22	Cys166, Thr167, Thr225, Arg249, Asn335
	Miltefosine	−4.5 ± 0.2	Pro136, Cys166	Thr199, Arg249
Pyruvate kinase * PDB ID: 1PKL	α-amiryn	−9.9 ± 0.1	Gln42, Leu74, Val76, Glu438, Phe463, Ile41	Ser439
	Episwertenol	−8.5 ± 0.1	Arg19, Arg348, Leu351	Arg19, Arg22
	Methyl commate A	−8.3 ± 0.2	Ile78, Ala347, Arg348, Glu438	Glu348, Ser439
	Amphotericin B	−9.1 ± 0.1	Arg22, Ile 41, Gly44, Val76, Glu438, Val440, Phe463, Lys467	Arg19, Ser46, Asn67, Asn432, Ser439, Glu462
	Miltefosine	−4.9 ± 0.2	Val76, Tyr18	Ser439
Triose phosphate isomerase * PDB: 1AMK	Episwertenol	−8.3 ± 0.2	Ile68, Ala69, Lys70, Phe74, Glu104, Ile108	Arg98, Glu104, Lys112
	Amphotericin B	−6.9 ± 0.2	Tyr101, Gly103, Thr105, Thr13, Gln133, Val169	Arg99, Thr100, Glu104, Gln146
	Miltefosine	−4.3 ± 0.2	Ile68, Phe74, Ala69, Ile108	Lys112, Arg98
Aldolase * PDB: 1EPX	α-amiryn	−8.6 ± 0.1	Ala41, Glu44, Leu121, Leu280, Ala312,	
	Methyl commate A	−8.3 ± 0.2	Asp43, Glu44, Cys84, Leu121, Leu280, Ala312, Arg313, Lys239	Asp43, Lys239, Ser286, Ala312, Arg313
	Episwertenol	−8.2 ± 0.2	Ala312, Arg313	Arg52
	Amphotericin B	−8.6 ± 0.1	Glu44, Ser45, Leu121, Pro123, Gly130, Gln132, Lys162, Leu280, Ala312	Asp43, Lys116, Lys156, Arg158, Glu199, Gly282
	Miltefosine	−5.4 ± 0.2	Leu280, Ala312, Arg313	Lys156, Lys116, Arg158
Transketolase * PDB: 1R9J	α-amiryn	−8.2 ± 0.1	Trp309, Val 55, Leu313, Phe 327, Val328, Met331, Arg332	
	Episwertenol	−8.2 ± 0.1	Phe104, Pro52, Pro107, Arg 332	Pro52, Asp53
	Methyl commate A	−8.6 ± 0.1	Phe104	Met331
	Amphotericin B	−9.6 ± 0.1		Arg96, Asp53, Arg57, Arg103, Phe104, Asp420, Ala421, Asp 423, His450
	Miltefosine	−5.0 ± 0.1	Phe104, Ile508	Val109, Arg103, Asp423
Arginase ** PDB: 4ITY	Episwertenol	−8.7 ± 0.1	Leu190, Val193, Leu201, His202, Ala208	Lys198
	α-amiryn	−8.5 ± 0.1	Val193, Lys198, Leu201, Ala208, Val331, Arg332, Trp369	Arg191
	Amphotericin B	−7.4 ± 0.2	Pro27, His28, Asn152, Ala192, Asp194, Lys198, Met211, Val259,	Arg191, Val193, Arg260, Gly261
	Miltefosine	−4.9 ± 0.2	Lys198, Leu190, Leu201, Val193, Ala208, Ala207	Ser210, his213
Cysteine peptidases A *** PDB: 2C34	Methyl commate A	−8.1 ± 0.1	Met35, Pro30, Asp45, Tyr58, Pro60, Pro95	Cys56, Asp45, Lys57
	Episwertenol	−8.0 ± 0.1	Pro30, Phe96	Tyr58
	Amphotericin B	−7.6 ± 0.2		Thr31, Gly69, Val68
	Miltefosine	−4.4 ± 0.2	Pro95, Met35, Phe96, Pro30, Pro60, Tyr58	

* Glycolysis biosynthesis pathway: GAPDH, pyruvate kinase, triose phosphate isomerase, aldolase, transketolase. ** Polyamine salvage pathway: Arginase. *** Proteinase pathway: Cysteine peptidase A. Each binding energy (kcal/mol) represents the mean ± standard error of the mean of six independent replicas kcal/mol^−1^.

## Data Availability

Not applicable.

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
