# Peer review of "Antileishmanial Activity and In Silico Molecular Docking Studies of Malachra alceifolia Jacq. Fractions against Leishmania mexicana Amastigotes"

_tropicalmed, 2023, doi:10.3390/tropicalmed8020115_

Round 1

Reviewer 1 Report

The study is significant as there is a high need of drug formulation against leishmaniasis.

The manuscript is  well written  however it is suggested to add more references in introduction section 

Author Response

The manuscript Antileishmanial activity and in silico molecular docking studies of Malachra alceifolia Jacq., fractions against Leishmania mexicana amastigotes were adjusted to support the hypothesis.

 Point 1: The manuscript is well written however it is suggested to add more references in introduction section.

Response 1: Further references are added in the introduction and in the full text.  Title adjusted for correlation of in vitro and in silico activity.

Reviewer 2 Report

-" W"

R:in the abstract What is mean?

-"which have serious side effects such as cardiotoxic, nephrotoxic and the development of insulin-dependent diabetes"

R:In the Introduction ,write a citation

-"challenge for many researchers. Studying proteases that play an important role in virulence factors in Leishmania spp., [11] involved in many parasitic activities, such as tissue invasion, survival in macrophages and modulation of the host immune response [12], contribute to the search for potential targets for treatment development [13]. "

R:Write the citation [11-13] in the final of paragraph.

-"pyruvate kinase, glyceraldehyde-3-phosphate dehydrogenase, arginase, triose phosphate isomerase aldolase, and phosphoglucose isomerase of L. mexicana, proteases as potential".

R: Describe the importante of these enzimes for the parasite.

-" The axenic amastigotes Leishmania mexicana pifanoi (MHOM/VE/60/Ltrod) ". How was obteined the axenic amastigote?

-Figure 4.

R: Describe the components

-Figure 2 b

R: Describe the abbreviations

-Figure 2b

R:  alpha-amyrin effects should be described. Specify the effects on axenic amastigote

Author Response

The manuscript Antileishmanial activity and in silico molecular docking studies of Malachra alceifolia Jacq., fractions against Leishmania mexicana amastigotes were adjusted to support the hypothesis.

Point 1: " W".

Response 1:. The “W” word was removed

Point 2: which have serious side effects such as cardiotoxic, nephrotoxic and the development of insulin-dependent diabetes". In the Introduction, write a citation

Response 2: The wording of the paragraph announcing the effect of the drugs is adjusted, the text is referenced.

Importantly, the treatment of leishmaniasis is complicated by the administration of pentavalent antimonial, liposomal amphotericin B, pentamidine, paromomycin and miltefosine. These drugs generally have good efficacy but which have serious side effects such as cardiotoxic, nephrotoxic, cytotoxicity and the development of insulin-dependent diabetes [10].  

Point 3: challenge for many researchers. Studying proteases that play an important role in virulence factors in Leishmania spp., [11] involved in many parasitic activities, such as tissue invasion, survival in macrophages and modulation of the host immune response [12], contribute to the search for potential targets for treatment development [13]. " R: Write the citation [11-13] in the final of paragraph.

Response 3: This paragraph was better worded and quoted at the end.

Currently, the development of new antileishmanial compounds less toxic, readily available and affordable for people affected by Leishmania has become a challenge for many researchers. The degradome of Leishmania spp., contains proteases with 18 clans classified into 35 families that play an important role in virulence factors in Leishmania spp., involved in many parasitic activities such as tissue invasion, survival in macrophages and modulation of the host immune response which contribute to the search for potential drug targets for treatment development [11–13].

Point 4: pyruvate kinase, glyceraldehyde-3-phosphate dehydrogenase, arginase, triose phosphate isomerase aldolase, and phosphoglucose isomerase of L. mexicana, proteases as potential". R: Describe the importance of these enzymes for the parasite.

Response 4: Described in the introduction and discussion.

Though, much remains to identify the active principles of medicinal plants as sources of drugs with potent leishmanicidal activity in inhibitors of enzymes essential for replication, cell cycle regulation or the production of virulence factors [15].

The proteins of biosynthesis of glycolysis, polyamine salvage and proteinase are involved in the steps of parasite invasion and migration inside the host, immune evasion, pathogenesis and disease outcome [49]. The L. mexicana protein targets: 1PKL the glycolytic processes; 1EPX glycolytic pathway, fructose-1,6-bisphosphate conversion to glyceraldehyde-3-phosphate, dihydroxyacetone phosphate; 1AMK plays a central role in the glycolysis process as a catalyst of dihydroxyacetone phosphate (DHAP), D-glyceraldehyde-3-phosphate (GAP); 1R9J transfers two-carbon glycolaldehyde units from ketose-donors to aldose-acceptor sugars; 1A7K, 1Q50 catalyzes oxidation-reduction reactions, defense against oxidative stress; 2C34 is central to host-parasite interactions, virulence factors; 4ITY, cellular growth, survival and  robust pathogenesis [15, 50–53]. The cysteine peptidases proteins play a crucial role in host cell-parasite interaction thus affecting host immune response, autophagy, cell death and growth suppressors. Glycolysis biosynthesis Leishmania depends on host for the carbon sources which are used for ATP generation in peroxisomes like cell organelles called glycosomes, these glycolytic proteins involved in this pathway are also potential drug targets [54].

Point 5: Los amastigotes axénicos Leishmania mexicana pifanoi (MHOM/VE/60/Ltrod) ". ¿Como se obtuvo el amastigoto axénico?

Response 5: The axenic amastigotes L. mexicana pifanoi (MHOM/VE/60/Ltrod) and RAW 264.7 murine macrophage cell line were obtained from Dr. Luis Rivas Centro de Investigaciones Biológicas Margarita Salas (C.S.I.C), Madrid, Spain

Described in Material and Methods, 2.4. In vitro anti-amastigote activity on L. mexicana pifanoi reference 21.

Protocolo. Luque-Ortega, J.R., Rivas, L. 2010. Characterization of the Leishmanicidal Activity of Antimicrobial Peptides. In: Giuliani, A., Rinaldi, A. (eds) Antimicrobial Peptides. Methods in Molecular Biology, vol 618. Humana Press, Totowa, NJ. https://doi.org/10.1007/978-1-60761-594-1_25.

Point 6: Figure 4. Describe the components, Figure 2b. Describe the abbreviations. alpha-amyrin effects should be described. Specify the effects on axenic amastigote

Response 6: The figure was removed and adjusted in supplementary material.  Currently, Figure 2b is part of the text of the in-silico study results. 

The study did not evaluate pure compounds, only active fractions with the prediction of possible compounds present in the fraction identified by GC-MS for in silico assays.

Reviewer 3 Report

Comments on tropicalmed-2168795:

The current paper reports a combined experimental and computational investigation of protein-ligand interactions involving L. mexicana. The topic could be suitable to the tropicalmed audience, but the presentation and scientific issues should be solved first. Below, I provide detailed comments.

The coloring regime in Figure 2 must be improved. Currently, the error bar in the bar plots cannot be clearly presented.

The authors claim that they have determined the binding affinities of compounds in their paper. However, Ive read the manuscript carefully but still have trouble finding the position that they report these numbers. The only property with the energy unit kcal/mol is the docking score, which is obviously not the experimental binding affinity.

Why kcal/mol is written as Kcal/mol in this paper?

Many critical details of the modelling are missing. For example, how do the authors construct the initial structures of ligands? All ligands under consideration are rather large and have a number of rotatable bonds, which makes it difficult to determine the most favorable conformation for each ligand. This initial setting could have a significant impact on the modelling outcome and must be provided. Other modelling details such as the parameters used in docking should also be provided.

The biggest problem concerning scientific rigor should be the application of molecular docking. The docking tool is widely applied in drug screening and molecular modelling, but is never considered a predictive instrument or a tool of reliability in any application. The applications in the drug industry are mainly on pre-screening a large set of potentially useful molecules, in order to provide a smaller set of hits. The reason that the screened compounds are called hits is rather obvious: they are very probable to do not have a satisfactory binding affinity, mainly due to the inaccuracy of the docking tool. Thus, I really question the motivation of using docking in this work to obtain the binding affinity (docking score).

The second aim of using docking calculations in this work is to secure a protein-ligand binding mode. However, this procedure is also questionable. Without extensive conformational search and also due to the inaccuracy of the scoring function, docking calculations cannot produce a good estimate of the binding mode. In many cases, they deviate significantly from the experimental mode. The authors should either perform more rigorous conformational search with atomistic simulations augmented by enhanced sampling techniques (see e.g., 10.1021/acs.jcim.1c01208 and 10.1016/j.carbpol.2022.120050) to obtain a reliable estimate of the binding mode, or add several paragraphs discussing the potential problems in their calculations and the resulting binding affinity estimates and binding modes. Without these modifications, the scientific rigor of the current work is questionable.

Even after reading the manuscript for several times, I cannot see why it has a 20-page landscape. Many results of little scientific importance are put in the main text, which makes the paper quite long. Normally, modern scientific publishing supports the use of the supporting material as a repository for technical data, figures and tables with relatively small scientific relevance. However, the authors obviously do not follow this common treatment.

According to the comments about the unreliability of molecular docking, Figure 4-5 and Table 5 should be moved to the supporting information due to their unjustified reliability. As an expert in molecular modelling, I consider these results literally useless. If the authors insist to include these docking results in their paper, they should at least move the figure to the supporting information to avoid overemphasizing the importance of these results.

The GC-MS analysis, which corresponds to Table 3 and Figure 1, does not have great importance either. They (at least the figure and table) should be moved to the supporting information.

The quality of Figure 4-5 should be improved. Currently, these plots seem rather non-satisfactory in published papers. The authors should read the author guideline of the journal in preparing these figures.

Author Response

The manuscript Antileishmanial activity and in silico molecular docking studies of Malachra alceifolia Jacq., fractions against Leishmania mexicana amastigotes were adjusted to support the hypothesis.

Point 1: The coloring regime in Figure 2 must be improved. Currently, the error bar in the bar plots cannot be clearly presented.

Response 1: Figures were improved

Figure 1a. The effect of fractions M. alceifolia on axenic amastigotes L. mexicana pifanoi survival. Parasites were incubated with different concentrations (2.5, 10.0 and 50.0 µg/mL) of chloroform fraction MA-IC and sub-fractions of MA-24F, with miltefosine (positive control 30 µM) and parasite untreated. The amastigote viability was determined by the MTT assay. Each bar represents the mean ± standard error of at least three independent experiments.  ***P < 0.05 with respect to the parasite untreated.

Figure 1b.  Cytotoxicity of fractions M. alceifolia on RAW 264.7 murine macrophage cell viability. RAW 264.7 were incubated with different concentrations (2.5, 10.0 and 50.0 µg/mL) of MA-IC and MA-24F fractions, cells untreated (Control), miltefosine (30 µM) and amphotericin B (20 µM) for 72 h. The cell viability was determined by the MTT assay. Each bar represents the mean ± standard error of at least three independent experiments.

Point 2: The authors claim that they have determined the binding affinities of compounds in their paper. However, I’ve read the manuscript carefully but still have trouble finding the position that they report these numbers. The only property with the energy unit kcal/mol is the docking score, which is obviously not the experimental binding affinity.

Why kcal/mol is written as Kcal/mol in this paper?

Response 2: It is adjusted in the methodology, description and results of the same in the molecular docking study for the interaction of possible allosteric sites between proteins-ligands (proteins of L. mexicana - compounds identified by CG-MS active fractions M. alceifolia). Adjusted throughout the text Kcal/mol by kcal/mol.

Point 3: The biggest problem concerning scientific rigor should be the application of molecular docking. The docking tool is widely applied in drug screening and molecular modelling, but is never considered a predictive instrument or a tool of reliability in any application. The applications in the drug industry are mainly on pre-screening a large set of potentially useful molecules, in order to provide a smaller set of hits. The reason that the screened compounds are called hits is rather obvious: they are very probable to do not have a satisfactory binding affinity, mainly due to the inaccuracy of the docking tool. Thus, I really question the motivation of using docking in this work to obtain the binding affinity (docking score).

Response 3: It is adjusted in the methodology, results description the same in the molecular docking study for the interaction of possible allosteric sites between proteins-ligands (proteins of L. mexicana - compounds identified by CG-MS active fractions M. alceifolia).

Point 4: The second aim of using docking calculations in this work is to secure a protein-ligand binding mode. However, this procedure is also questionable. Without extensive conformational search and also due to the inaccuracy of the scoring function, docking calculations cannot produce a good estimate of the binding mode. In many cases, they deviate significantly from the experimental mode. The authors should either perform more rigorous conformational search with atomistic simulations augmented by enhanced sampling techniques (see e.g., 10.1021/acs.jcim.1c01208 and 10.1016/j.carbpol.2022.120050) to obtain a reliable estimate of the binding mode, or add several paragraphs discussing the potential problems in their calculations and the resulting binding affinity estimates and binding modes. Without these modifications, the scientific rigor of the current work is questionable.

Response 4: I t is adjusted in the methodology, results description the same in the molecular docking study for the interaction of possible allosteric sites between proteins-ligands (proteins of L. mexicana - compounds identified by CG-MS active fractions M. alceifolia).

Table 1. Binding affinity estimation of energies and molecular interactions of phytocomponents identified in the Ma-24F fraction of leaves of M. alceifolia by GC-MS upon docking in a potential allosteric sites of significant L. mexicana target proteins.

TS3. Binding affinity estimation energies and molecular interactions of phytocomponents identified in the Ma-24F fraction of leaves of M. alceifolia by GC-MS upon docking in a potential allosteric site of significant L. mexicana target proteins.

Point 5: Even after reading the manuscript for several times, I cannot see why it has a 20-page landscape. Many results of little scientific importance are put in the main text, which makes the paper quite long. Normally, modern scientific publishing supports the use of the supporting material as a repository for technical data, figures and tables with relatively small scientific relevance. However, the authors obviously do not follow this common treatment.

Response 5: The full text is adjusted and included in the supplementary material.

Point 6: According to the comments about the unreliability of molecular docking, Figure 4-5 and Table 5 should be moved to the supporting information due to their unjustified reliability. As an expert in molecular modelling, I consider these results literally useless. If the authors insist to include these docking results in their paper, they should at least move the figure to the supporting information to avoid overemphasizing the importance of these results.

Response 6: The results of the figures are adjusted:

Figure 2a. Episwertenol compound interaction of allosteric binding sites of target proteins pyruvate kinase (PDB ID: 1PKL), GAPDH (PDB ID: 1A7K), triose phosphate isomerase (PDB ID: 1AMK), aldolase (PDB ID: 1EPX), transketolase (PDB: 1R9J), arginase (PDB: 4ITY) and Cysteine peptidase A (PDB: 2C34).

Figure 2b. α-Amiryn compound interaction of allosteric binding sites of target proteins Pyruvate kinase (PDB ID: 1PKL), GAPDH (PDB ID: 1A7K), aldolase (PDB ID: 1EPX), transketolase (PDB: 1R9J) and arginase (PDB: 4ITY).

Figure 2c. Methyl commate A compound interaction of allosteric binding sites of target proteins Pyruvate kinase (PDB ID: 1PKL), aldolase (PDB ID: 1EPX), transketolase (PDB: 1R9J) and Cysteine peptidase A (PDB: 2C34).

Other data in supplementary materials

Point 7: The GC-MS analysis, which corresponds to Table 3 and Figure 1, does not have great importance either. They (at least the figure and table) should be moved to the supporting information.

Reviewer 4 Report

In the present manuscript, authors have explored " In silico molecular docking studies and antileishmanial activity of fractions Malachra alceifolia Jacq., against Leishmania mexicana proteases". The study lacks a specific hypothesis being tested.

1. Introduction to be revised properly, why the authors have chosen the plant species. 2. Molecular dynamics simulations are usually used to address protein folding issues or protein–ligand complex stability through energy profile analysis over time which is required to perform for the test-protein complex, standard-protein complex, and apoprotein at least for 100ns.

3. In Table 5 the value of binding energy should be in decimals not in commas Eg: α-Amiryn -9,9 it should be like this -9.9

4. The discussion needs to be revised properly by correlating all the data from in silico and in vitro studies.

5.  There is no discussion about the Pharmacokinetic properties of the selected ligands in comparison with standard drugs.

6. Very Poor image qualities must be improved

7. 3D interactions of amino acids should be added.

8. Typo errors to be rectified. 

Author Response

The manuscript Antileishmanial activity and in silico molecular docking studies of Malachra alceifolia Jacq., fractions against Leishmania mexicana amastigotes were adjusted to support the hypothesis.

Point 1: In the present manuscript, authors have explored " In silico molecular docking studies and antileishmanial activity of fractions Malachra alceifolia Jacq., against Leishmania mexicana proteases". The study lacks a specific hypothesis being tested.

Response 1: The manuscript Antileishmanial activity and in silico molecular docking studies of Malachra alceifolia Jacq. fractions against Leishmania mexicana amastigotes were adjusted to support the hypothesis.

Point 2: Introduction to be revised properly, why the authors have chosen the plant species. 2. Molecular dynamics simulations are usually used to address protein folding issues or protein–ligand complex stability through energy profile analysis over time which is required to perform for the test-protein complex, standard-protein complex, and apoprotein at least for 100ns.

Response 2: It is adjusted in the methodology, description and results of the same in the molecular docking study for the interaction of possible allosteric sites between proteins-ligands (proteins of L. mexicana - compounds identified by CG-MS active fractions M. alceifolia).

Point 3: In Table 5 the value of binding energy should be in decimals not in commas Eg: α-Amiryn -9,9 it should be like this -9.9

Response 3: Table 5 was adjusted by Table 1. Binding affinity estimation of energies and molecular interactions of phytocomponents identified in the Ma-24F fraction of leaves of M. alceifolia by GC-MS upon docking in a potential allosteric sites of significant L. mexicana target proteins.

Supplementary Materials TS3. Binding affinity estimation energies and molecular interactions of phytocomponents identified in the Ma-24F fraction of leaves of M. alceifolia by GC-MS upon docking in a potential allosteric site of significant L. mexicana target proteins.

Point 4: The discussion needs to be revised properly by correlating all the data from in silico and in vitro studies.

Response 4: The discussion in the manuscript was adjusted by appending the correlation of in vitro with in silico.

Point 5: There is no discussion about the Pharmacokinetic properties of the selected ligands in comparison with standard drugs.

Response 5: The pharmacological profiling of the fifteen compounds with high binding affinities with the L. mexicana protein was not evaluated.

Point 6: Very Poor image qualities must be improved

Response 6: The results of the figures are adjusted:

Figure 2a. Episwertenol compound interaction of allosteric binding sites of target proteins pyruvate kinase (PDB ID: 1PKL), GAPDH (PDB ID: 1A7K), triose phosphate isomerase (PDB ID: 1AMK), aldolase (PDB ID: 1EPX), transketolase (PDB: 1R9J), arginase (PDB: 4ITY) and Cysteine peptidase A (PDB: 2C34).

Figure 2b. α-Amiryn compound interaction of allosteric binding sites of target proteins Pyruvate kinase (PDB ID: 1PKL), GAPDH (PDB ID: 1A7K), aldolase (PDB ID: 1EPX), transketolase (PDB: 1R9J) and arginase (PDB: 4ITY).

Figure 2c. Methyl commate A compound interaction of allosteric binding sites of target proteins Pyruvate kinase (PDB ID: 1PKL), aldolase (PDB ID: 1EPX), transketolase (PDB: 1R9J) and Cysteine peptidase A (PDB: 2C34).

Other data in supplementary materials

Point 7: 3D interactions of amino acids should be added.

Response 7: The 3D interactions of amino acids are adjusted in the graphs.

Point 8: Typo errors to be rectified.

Response 8: Typos in the manuscript were corrected.

Round 2

Reviewer 3 Report

Comments on tropicalmed-2168795.R1:

The submitted first revision of the manuscript seems still below the standard, despite the minor and careless modifications by the authors. Presentation issues and scientifically relevant problems keep degrading the readability and the reliability of the paper. The quality of computational results via the crude molecular docking is acknowledged to be low, but the docking outcomes are actually the core of the paper. Currently neither academic nor industrial applications would really rely on such an inaccurate method to draw conclusive recommendations from the computational perspective, and it is useless to have a scientific contribution presenting only docking predictions due to the well-known unreliability. The authors also have trouble providing a point-to-point response and modify their papers according to suggestions. Many of my comments are not properly responded or simply neglected. For example, although I have commented that many modelling details are missing, e.g., the initial structure of ligands due to the large number of rotatable bonds, in the revised manuscript many mentioned modelling details are still not provided. For such a careless revision, I believe that none of international journals with reputations would consider it as a properly responded response or carefully revised revision. Overall, for a scientific contribution of such a low quality, I cannot recommend publication in any international journal with reputations.  

Author Response

The manuscript Antileishmanial activity and in silico molecular docking studies of Malachra alceifolia Jacq., fractions against Leishmania mexicana amastigotes were adjusted to support the hypothesis.

Point 1: The submitted first revision of the manuscript seems still below the standard, despite the minor and careless modifications by the authors. Presentation issues and scientifically relevant problems keep degrading the readability and the reliability of the paper. The quality of computational results via the crude molecular docking is acknowledged to be low, but the docking outcomes are actually the core of the paper. Currently neither academic nor industrial applications would really rely on such an inaccurate method to draw conclusive recommendations from the computational perspective, and it is useless to have a scientific contribution presenting only docking predictions due to the well-known unreliability. The authors also have trouble providing a point-to-point response and modify their papers according to suggestions. Many of my comments are not properly responded or simply neglected. For example, although I have commented that many modelling details are missing, e.g., the initial structure of ligands due to the large number of rotatable bonds, in the revised manuscript many mentioned modelling details are still not provided. For such a careless revision, I believe that none of international journals with reputations would consider it as a properly responded response or carefully revised revision. Overall, for a scientific contribution of such a low quality, I cannot recommend publication in any international journal with reputations.

Point 1: The submitted first revision of the manuscript seems still below the standard, despite the minor and careless modifications by the authors.

Response 1: The authors introduced modifications in the first version of the manuscript to adjust to the reviewer's observations without altering the hypothesis or the object of the in vitro study of the antileishmanial activity of the active fractions of M. alceifolia and of the in silico study on the theoretical interaction of affinity energy in the allosteric binding site of target proteins of L. mexicana with these compounds. The reviewer's comments helped improve the structure and strengthen the manuscript.

Point 2: Presentation issues and scientifically relevant problems keep degrading the readability and the reliability of the paper. The quality of computational results via the crude molecular docking is acknowledged to be low, but the docking outcomes are actually the core of the paper.

Response 2: For he authors, this research starts from contextualizing empirical knowledge of the use of the leaves of this plant to provide solutions to health problems, especially infectious diseases, starting from obtaining extracts and fractions with potential antileishmanial activity in vitro, reinforced with the prediction through computational tools in the search for possible antileishmanial phytocompounds against allosteric domains in proteins of L. mexicana. According to the reviewer's observations, the methodology is improved and the results of the computational studies are organized, providing a baseline to continue with in vitro studies that continue validating the pharmacological potential of M. alceifolia compounds against Leishmania sp. amastigotes.

Point 3: Currently neither academic nor industrial applications would really rely on such an inaccurate method to draw conclusive recommendations from the computational perspective, and it is useless to have a scientific contribution presenting only docking predictions due to the well-known unreliability.

Response 3: The authors highlight the computational study as a baseline tool to further strengthen in vitro and in vivo studies of antileishmanial activity with pure compounds obtained from M. alceifolia, an underreported plant with great ethnobotanical and ethnopharmacological value.

The reviewer's observations allowed for the organization of the methodology and the results of the computational studies, giving possible approaches to the search for phytocompounds with antileishmanial potential.

Point 4: The authors also have trouble providing a point-to-point response and modify their papers according to suggestions. Many of my comments are not properly responded or simply neglected. For example, although I have commented that many modelling details are missing, e.g., the initial structure of ligands due to the large number of rotatable bonds, in the revised manuscript many mentioned modelling details are still not provided.

Response 4: The authors expanded their response in the last review about their observations, reviewing in detail the computational study methodology:

  • The prediction of allosteric binding sites
  • Virtual Screening
  • Characterization of mechanism of allosteric binding sites.
  • Supplementary Materials: TS3. Scoring Function estimation of binding affinity energy and molecular interactions of phytocomponents identified in the Ma-24F fraction of leaves of alceifolia by GC-MS upon docking in a potential allosteric sites of significant L. mexicana target proteins for AutoDock Vina program and validated values using the CB-Dock program.
  • Supplementary Materials TS4. Identification of amino acid residues in the allosteric site of proteins in L. mexicana selected for docking.

The authors thank the reviewer for all his contributions to improving the quality of the manuscript, highlighting his contribution to the importance of publishing studies that generate impact and contribute to science.

Reviewer 4 Report

The manuscript can be accepted in the present form.

Author Response

The manuscript Antileishmanial activity and in silico molecular docking studies of Malachra alceifolia Jacq., fractions against Leishmania mexicana amastigotes were adjusted to support the hypothesis.

 Point 1: The manuscript can be accepted in the present form.

The authors appreciate the opportunity, support and quality of the reviewers' comments to submit this manuscript for publication in your Journal of Tropical Medicine and Infectious Diseases.

Round 3

Reviewer 3 Report

 .